# FlareX: A Physics-Informed Dataset for Lens Flare Removal via 2D Synthesis and 3D Rendering

**Lishen Qu**[1,3], **Zhihao Liu**[3], **Jinshan Pan**[4],
**Shihao Zhou**[1,3], **Jinglei Shi**[3,5], **Duosheng Chen**[3], **Jufeng Yang**[1,2,3,*]

[1]Nankai International Advanced Research Institute (SHENZHEN·FUTIAN)
[2]Peng Cheng Laboratory    [3]College of Computer Science, Nankai University
[4]Nanjing University of Science and Technology
[5]Key Lab of SCCI, Dalian University of Technology

{qulishen, 2212602, zhoushihao96, duoshengchen}@mail.nankai.edu.cn
jspan@njust.edu.cn, {jinglei.shi, yangjufeng}@nankai.edu.cn
https://github.com/qulishen/FlareX

## Abstract

Lens flare occurs when shooting towards strong light sources, significantly degrading the visual quality of images. Due to the difficulty in capturing flare-corrupted and flare-free image pairs in the real world, existing datasets are typically synthesized in 2D by overlaying artificial flare templates onto background images. However, the lack of flare diversity in templates and the neglect of physical principles in the synthesis process hinder models trained on these datasets from generalizing well to real-world scenarios. To address these challenges, we propose a new physics-informed method for flare data generation, which consists of three stages: parameterized template creation, the laws of illumination-aware 2D synthesis, and physical engine-based 3D rendering, which finally gives us a mi**X**ed flare dataset that incorporates both 2D and 3D perspectives, namely **FlareX**. This dataset offers 9,500 2D templates derived from 95 flare patterns and 3,000 flare image pairs rendered from 60 3D scenes. Furthermore, we design a masking approach to obtain real-world flare-free images from their corrupted counterparts to measure the performance of the model on real-world images. Extensive experiments demonstrate the effectiveness of our method and dataset.

## 1 Introduction

Lens flare often appears when capturing images against strong light, due to light scattering and reflection within the lens system [1, 2]. This phenomenon leads to color shifts and loss of information [3, 4], resulting in degraded image quality and lowering the performance of downstream vision tasks [5, 6, 7]. Depending on various lens designs, as shown in Figure 1(a) lighting conditions, flares can differ greatly in color, shape, intensity, and spatial extent as depicted in Figure 1(b), making flare removal a challenging task.

Traditional methods [8, 9] add an anti-reflective coating to the lenses to reduce reflective flares. However, the coating's high cost and limited effectiveness restrict its widespread adoption in consumer-grade devices. To benefit from lower costs and stronger flare removal capability, researchers [10, 11, 12] begin to remove flare through deep learning algorithms [13, 14, 15, 16, 17, 18, 19, 20], driven by the large dataset [3, 21]. Besides, it is challenging to obtain a large number of paired flare images through shooting. Cleaning the lens may suppress scattering flare, but cannot eliminate reflective flare caused by lens internal imperfections [22] and intense flare.

---

[*]Corresponding Author.

39th Conference on Neural Information Processing Systems (NeurIPS 2025) Track on Datasets and Benchmarks.

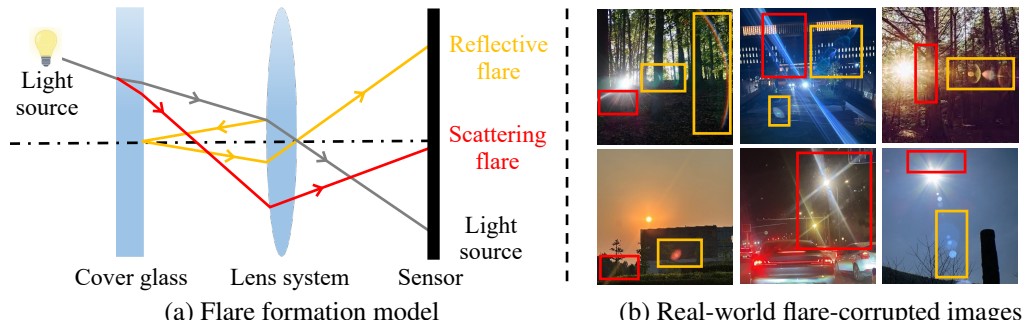

|  |  |
|---|---|
| (a) Flare formation model | (b) Real-world flare-corrupted images |

Figure 1: Flare formation mechanisms and real-world flare-corrupted images. (a) The red and yellow lines respectively represent light rays that create scattering and reflective flares. (b) We use the same colors as (a) to highlight the corresponding flares. Under various lighting conditions and lens designs (e.g., aperture size, lens material), flares appear in a wide range of colors and shapes (e.g., circles, polygons, and streaks).

Given the difficulty of obtaining paired flare images in the real world, previous works [1, 2, 23] rely on simulations. Wu *et al.* [24] and Dai *et al.* [3] build datasets of flare templates using Spread Point Function and Adobe After Effects, respectively. Then, they added these templates to the background image [25] to synthesize flare-corrupted images. These datasets [24, 3] include flare patterns caused by point light sources but lack coverage of real-world scenarios with multiple reflections and non-spherical light sources. Besides, their synthesis process neglects the physical properties of flares. For example, the overall brightness of the flare is closely correlated with the distance between the light source and the camera. As a result, models trained on these datasets have limited effectiveness in coping with various types of flares in the real world.

To train a robust neural network model, it is necessary to have a realistic dataset with rich flare patterns. In this paper, we present a data generation framework based on a 3D physical engine Blender, which consists of three stages, as shown in Figure 2. First, to better simulate real-world lens flares, we parameterize key factors, such as light intensity, the times of reflection, and glass pollution. This enables us to generate 9,500 flare templates derived from 95 types of flare, covering a wider range of patterns in the real world, compared to previous datasets [24, 3]. Second, in the synthesis process, we incorporate *the laws of illumination* [26, 27, 28] to establish the relationship between the intensity of the flare and the spatial position of the light source. With the help of an estimated depth map, the intensity of flares in the synthetic images appears more realistic than random addition. Finally, we construct 60 3D scenes by customizing the placement of flares in the appropriate place instead of adding them randomly. Leveraging the 3D physics engine, we render 3,000 flare image pairs that inherently follow physical laws. By mixing 2D synthetic image pairs derived from templates and rendered images from various perspectives in 3D scenes, we introduce a physics-informed dataset called FlareX, which can support future research in flare removal. In addition, due to existing methods [24, 3] that struggle to capture the image without flares, we propose a masking approach to evaluate the model's performance in the real world. In this way, we collect 100 image pairs with a resolution of $3024 \times 3024$, containing various types of flare for testing.

Our main contributions can be summarized as follows: (i) To address the limitations of flare pattern diversity in existing datasets, we create 9,500 templates derived from 95 types of flare with different parameter settings. (ii) We improve the existing 2D synthesis pipeline by incorporating the laws of illumination. And we further build 60 3D scenes with flares to render 3,000 image pairs as complements to build a physical-realistic dataset. (iii) We propose a masking approach to obtain flare-free images from flare-corrupted ones for better model assessment, and carry out extensive experiments to demonstrate the effectiveness of our method and dataset.

## 2 Related work

**Flare training dataset.** Due to various constraints and variables, collecting a large-scale dataset of paired flare images in the real world is an extremely challenging task [30]. Wu *et al.* [24] divide a flare-corrupted image into a flare template and a background, thereby building the first semi-synthetic flare dataset, which includes 2,001 captured flare templates and 3,000 simulated flare templates.

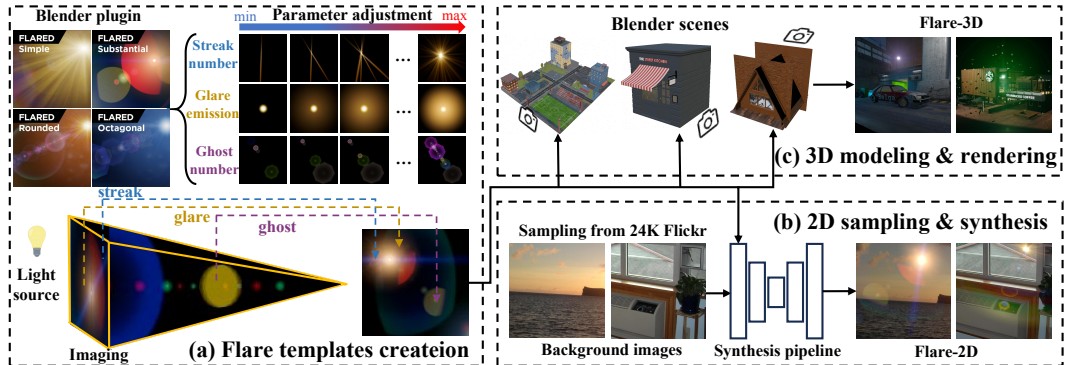

Figure 2: Illustration of our flare dataset generation framework. (a) The raw material of flares comes from the Blender plugin [29]. We manually create flare templates and adjust both flare and camera parameters to generate a wide variety of templates. (b) The synthesis pipeline generates flare-corr upted images by adding flare templates to background images sampled from the 24K Flickr image dataset [25]. (c) The 3D rendering approach constructs scenes with flare and renders them from various camera perspectives.

However, given the insufficient variety of Wu's dataset and inadequate consideration of nighttime conditions, Dai *et al.* [3] use the plugin in Adobe After Effects (AE) to create nighttime flares, including 5,000 scattering flare templates and 2,000 reflective flare templates, called Flare7K dataset. Then, Dai *et al.* capture 962 flare templates using three different cameras, expanding this collection into the Flare7K++ dataset [21]. Recently, Florin *et al.* [31] develop a synthetic flare dataset named SDFRD, specifically targeting digital single-lens reflex cameras.

**Flare-corrupted image generation.** The current methods randomly add the flare templates of [24, 3, 21] to the background images which are sampled from the 24K Flickr [25]. Wu *et al.* [24] and Dai *et al.* [3] apply random affine transformations to flare templates without considering physical principles, resulting in noticeable unrealism (e.g., excessive brightness or oversized flares). Zhou *et al.* [32] argue that simple addition and numerical truncation can cause overflow and distribution shift, leading them to propose an improved synthesis method. Jin *et al.* [33] demonstrate that lens flares captured by the same imaging system tend to be similar. Consequently, they select the same flare templates when randomly adding multiple flares which yields better results. To address the issue of unrealistic brightness, we improve the method of overlaying flares onto background images by introducing the laws of illumination. To position flares in more appropriate locations instead of placing them randomly, we further render 3D scenes to generate the remainder of the dataset.

**Flare test dataset.** Wu *et al.* [24] provide a test set comprising only 20 real image pairs, making it insufficient for evaluating performance in the real world. Dai *et al.* [3] extend this by introducing a nighttime test set containing 100 real captured flare image pairs. However, both datasets suffer from relatively low resolution (typically $512 \times 512$) and mainly contain simple and small-scale flare patterns. FlareReal600 [34] improves the resolution by providing 50 image pairs at 2K, but the ground truth is still acquired by cleaning the lens, which cannot eliminate internal reflective and severe flares. Zhou *et al.* [32] address the device diversity issue by collecting a test set using 10 mobile phones, but their dataset lacks ground truth altogether, making quantitative evaluation impossible. To tackle the difficulty of capturing reflective flare image pairs in real-world conditions, BracketFlare [22] constructs the first synthetic dataset specifically for reflective flares. Overall, current flare test datasets are limited either by resolution, diversity of flare types, or the lack of reliable ground truth, motivating the need for a more comprehensive evaluation benchmark.

## 3   FlareX dataset

Our dataset creation process consists of three stages: creating flare templates, 2D synthesis using the templates, and directly rendering 3D scenes. By incorporating the laws of illumination into the 2D synthesis process, we generate a large number of flare images with realistic intensity. Furthermore,

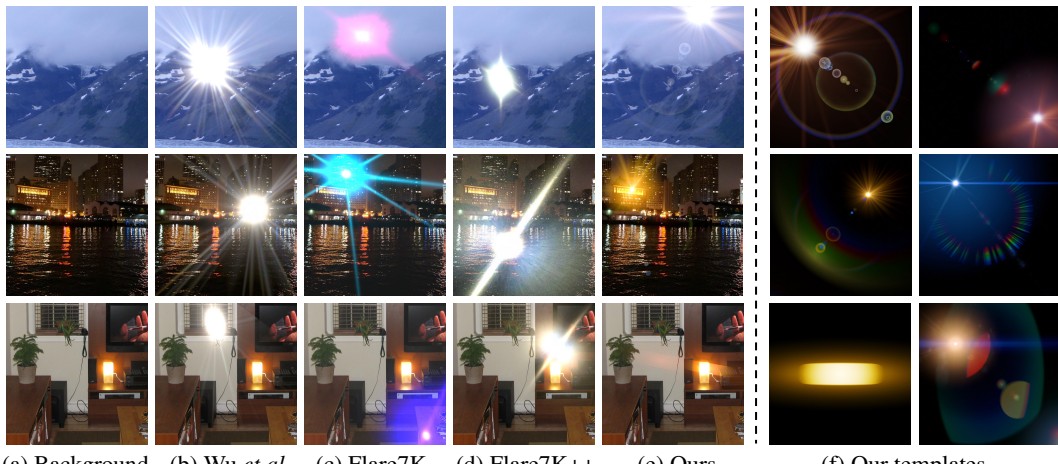

(a) Background    (b) Wu *et al.*    (c) Flare7K    (d) Flare7K++    (e) Ours      (f) Our templates

Figure 3: Comparison of synthetic flare-corrupted images and examples of our flare templates. Featuring various light sources and flares in different colors and shapes, our flare templates are capable of simulating sun halo effects in daytime and flares produced by non-spherical light sources. Note that Flare7K++ [21] specifically refers to the additional flare templates beyond those in Flare7K [3].

Table 1: Comparison of existing flare datasets. "✗" and "✓" indicate whether the dataset has the property. Note that the dataset of Wu *et al.* [24] only contains daytime flares while Flare7K [3] and FlareReal600 [34] only include nighttime flares.

| Dataset | Type | Number | Multiple reflection | Light source annotation | Day | Night |
|---|---|---|---|---|---|---|
| Wu *et al.* [24] | 3 | 5001 | ✗ | ✗ | ✓ | ✗ |
| Flare7K [3] | 35 | 7000 | ✗ | ✓ | ✗ | ✓ |
| FlareReal600 [34] | - | 600 | ✗ | ✗ | ✗ | ✓ |
| Flare-R [21] | - | 962 | ✗ | ✓ | ✗ | ✓ |
| SDFRD [35] | 3 | - | ✗ | ✗ | ✓ | ✓ |
| FlareX (Ours) | 95 | 9500+3000 | ✓ | ✓ | ✓ | ✓ |

3D scene rendering naturally adheres to physical laws, and we can place flares in locations where they are more likely to appear, rather than randomly adding them to backgrounds. To leverage the strengths of both 2D synthesis and 3D rendering, we mix the data generated by them to propose a physics-informed flare dataset, namely FlareX. We compare our dataset with existing datasets [24, 3, 34] in Table 1. Our dataset offers 9,500 flare templates derived from 95 types of flares to synthesize image pairs, along with 3,000 pairs of 3D-rendered images, featuring diverse patterns and covering both daytime and nighttime scenarios. For ease of reference, we abbreviate the two parts of FlareX as Flare-2D and Flare-3D in the following text.

## 3.1 Flare template creation

The existing datasets [3, 24] lack diversity and find it challenging to physically simulate lens flare, as this requires computing mutual constraints between flare components, which is a highly complex task. To solve the issues of diversity and unrealism, we use the 3D graphics engine Blender to simulate flares in alignment with physical laws.

First, we create a flare that includes multiple components, such as light source, streak, iris, and glare. We utilize the flare patterns from the Flared plugin to avoid creating flares from scratch. We manually adjust the camera focal length and various parameters of these flare components, including position, color, size, shape, etc. Most importantly, we bind the light source to a spatial point and apply the mutual constraints preset by Blender [29] among flare components, allowing the movement of the light source to produce pattern changes that more closely resemble real-world behavior. We set the background to black and allow Blender rendering in flat to produce the flare templates. After removing all flare components except the light source, we re-render the scene to obtain the corresponding light source templates. This process allows us to generate annotations for all flare components. Referring

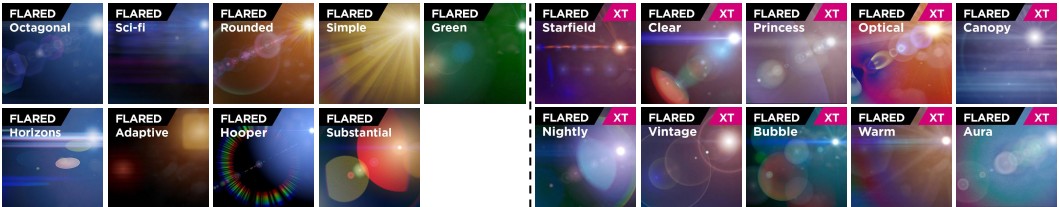

Figure 4: 19 categories of flare templates. The 9 categories on the left represent the basic flare types, while those on the right are more complex types, officially referred to as "XT".

to real-world flares, we create a total of 95 flare types using components in Figure 4, with each type producing 100 templates by adjusting parameters, resulting in 9,500 flare templates.

In Figure 3, we present synthetic images derived from our flare templates and those of previous works [24, 3, 21], showing scenes in daylight, nighttime, and indoor settings, respectively. Templates in previous datasets feature limited flare patterns, making them ineffective for synthesizing images across diverse scenes. In both daylight and nighttime scenes, our flare templates, with multi-reflection properties and controllable parameters, create more realistic effects that naturally integrate into the environment. For the indoor scene, we show the ability of our templates to simulate flares caused by the non-spherical light source.

### 3.2 Flare-2D synthesis

As existing data synthesis methods [24, 3] ignore the relationships between the appearance of lens flare and their spatial position, we improve them by introducing the laws of illumination [26, 27]. Different from existing methods, our data synthesis method can synthesize flare-corrupted images whose flare intensity adheres to physical laws. Our synthesis pipeline is illustrated in Figure 5.

First, we perform multiple random affine transformations of the flare separately and estimate the depth map of the background image using a pre-trained monocular depth estimation model [36]. Second, we develop a Brightness Adjustment Module (BAM) to adjust the brightness of flares that have been affine transformed. The above operations can be expressed by:

$$\mathbf{F}_{\mathbf{i}}'' = BAM(\mathcal{T}_i(\mathbf{F}), \mathcal{D}(\mathbf{B})) = BAM(\mathbf{F}_{\mathbf{i}}', \mathbf{D}), \tag{1}$$

where $\mathcal{T}_i$ and $\mathcal{D}$ denote the $i^{th}$ Random Affine Transformation and Depth Estimation, respectively. $\mathbf{F}$ denotes the origin flare image. $\mathbf{F}_{\mathbf{i}}'$ is the flare image after the $i^{th}$ random affine transformation. $\mathbf{F}_{\mathbf{i}}''$ is the flare image after brightness adjustment. $\mathbf{B}$ represents the background image and $\mathbf{D}$ represents its depth map.

Finally, we obtain multiple flares after adjusting the brightness, then add them to the background image to synthesize the final image with flares, which can be represented as:

$$\mathbf{I}_{\text{sys}} = Clip(\mathbf{B} + \sum_{i=1}^{n} \mathbf{F}_{\mathbf{i}}''), \tag{2}$$

where $Clip(\cdot)$ denotes clipping the addition to the range of $[0, 1]$, and $n$ represents the number of flares to be generated.

**Brightness adjustment module (BAM).** Intuitively, the closer the light source is to the lens, the more intense the flare appears. The laws of illumination [26] allow for a quantitative portrayal of this physical phenomenon and the formula is as follows:

$$E = \frac{I \cdot \cos \theta}{d^2}, \tag{3}$$

where $E$ indicates the illumination at a point on the plane, and $I$ is the luminous intensity of the light source. $\theta$ is the angle between the optical axis and the incident light rays, and $d$ is the distance from the light source to the illuminated point. To simplify explanations, we use the terms "angle of incidence" and "depth" in the rest of the paper.

We first perform Spatial Position Estimation (SPE) using the depth map and affine-transformed flare images to obtain the depth $d_i$ and the angle of incidence $\theta_i$ for each affine-transformed flare image.

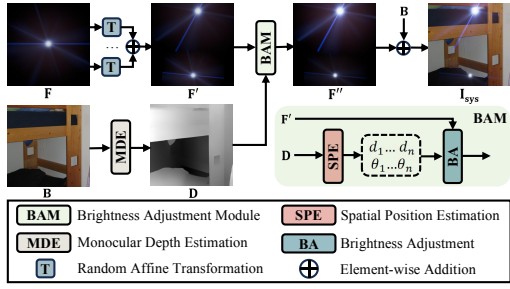

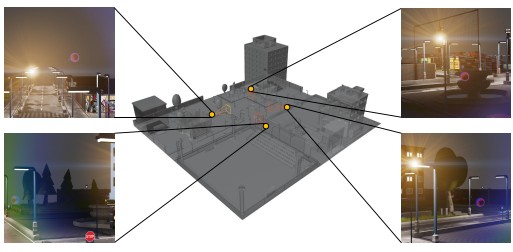

| | |
|---|---|
| **BAM** Brightness Adjustment Module | **SPE** Spatial Position Estimation |
| **MDE** Monocular Depth Estimation | **BA** Brightness Adjustment |
| **T** Random Affine Transformation | ⊕ Element-wise Addition |

Figure 5: Our synthesis pipeline. The flare-corrupted images generated based on the laws of illumination appear more realistic.

Figure 6: The 3D scene and the flare-corrupted images rendered from different camera positions. Due to the physical constraints, the variations in the shape of the flare and its position in the image align with physical laws.

In SPE, we utilize the average depth of all pixel points of the light source as the real distance between that light source and the lens, as expressed by the following equation:

$$d_i = \frac{1}{N} \sum_{j=1}^{N} \mathbf{D}_{(x_j, y_j)} \ , \ (x_j, y_j) \in \mathbf{L_i}, \tag{4}$$

where $\mathbf{L_i}$ symbolizes the light source area of the affine-transformed flare and $N$ represents the total number of pixel points of the light source. $x_j$ and $y_j$ denote the position of the $j^{th}$ pixel point.

For the calculation of the incident angle, since the camera parameters used for taking photos are unknown, we use the horizontal field of view to estimate. According to the law of similar triangles, we can obtain the following formula:

$$\theta_i = \arctan(\frac{2r_i}{W} \cdot \tan \frac{\varphi}{2}), \tag{5}$$

where $\varphi$ represents the horizontal field of view, $W$ denotes the width of the background image, and $r_i$ denotes the average distance from the pixel points of the $i^{th}$ light source to the center of the image.

Compared to the scenes being captured, the size of the lens can be ignored and represented as a point. After obtaining the depth and the incident angle, we substitute these values into Equation (3) to calculate the illumination of the lens from different light sources. The formula for making the final brightness adjustment is as follows:

$$\mathbf{F_i''} = \mathbf{F_i'} \cdot \frac{E_i}{\frac{1}{n} \cdot \sum_{i=1}^{n} E_i} = \frac{\mathbf{F_i'} \cdot \bar{d}^2}{d_i^2 \cdot \sqrt{1 + (\frac{2r_i}{W} \cdot \tan(\frac{\varphi}{2})^2)}} \ , \tag{6}$$

where $\bar{d}$ is the average depth of the image. Specifically, we use the light with a $0°$ incident angle and $\bar{d}$ as a reference for adjusting the brightness of each flare. Last but not least, due to variations in the fields of view among different cameras, training various models with the same dataset may lead to poor robustness. Compared to the previous synthesis method, our method can synthesize a dataset for a specific camera by adjusting the field of view $\varphi$.

## 3.3 Flare-3D construction

Since the estimated depth map is not always accurate, we propose constructing flare scenarios and rendering them directly to establish more precise physical constraints. This approach naturally adheres to physical laws and produces flare-corrupted images with a more realistic intensity and appearance. The existing synthesis operations may cause image distortion due to overflow [32], which can also be mitigated by the 3D modeling method.

First, we construct a 3D scene in Blender and add lens flare into the scene. Unlike the random affine transformation method, 3D rendering allows us to customize flare placement, positioning it in areas where flares are more likely to appear, such as near light sources, in the sky, and outside windows. Next, we keyframe the camera's movement path, enabling it to follow a trajectory within the scene

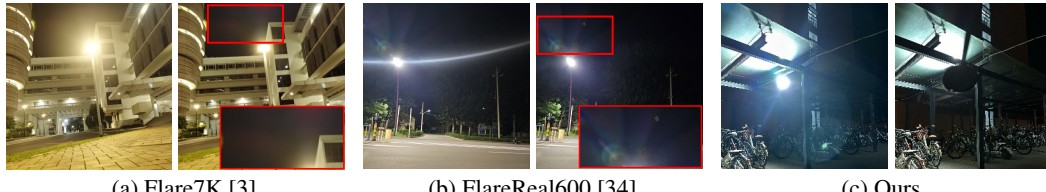

| (a) Flare7K [3] | (b) FlareReal600 [34] | (c) Ours |

Figure 7: Image pairs obtained through different collection methods. The ground truths in existing test sets may retain flares, especially when capturing strong light sources.

Table 2: Quantitative results and user study evaluation. Our test set better captures the performance improvement from Flare7K [3] to Flare7K++ [21], aligning more closely with human perceptual preferences.

| Dataset | Test on Flare7K [3] | | Test on FlareReal600 [34] | | Test on ours | |
|---|---|---|---|---|---|---|
| | PSNR | User study | PSNR | User study | PSNR | User study |
| Trained on Flare7K [3] | 33% | 4% | 34% | 8% | 7% | 5% |
| Trained on Flare7K++ [21] | 67% | 96% | 66% | 92% | 93% | 95% |

and capture images from various perspectives, as shown in Figure 6. Compared to 2D synthetic images, the flares in rendered images exhibit different appearances related to their spatial position, making them closer to real-world images. Finally, we move the camera along the predefined path to capture flare-corrupted images and repeat the process after removing the flares to capture flare-free ones. We construct 60 scenes with different flares and render 3,000 image pairs to develop Flare-3D.

Table 3: Quantitative comparison of various image restoration models on the proposed test set. We train the same models on five datasets, and the best and second-best scores are in **bold** and underlined.

| Dataset | HINet [37] | | | MPRNet [38] | | | Uformer [39] | | | Restormer [40] | | |
|---|---|---|---|---|---|---|---|---|---|---|---|---|
| | PSNR↑ | SSIM↑ | LPIPS↓ | PSNR↑ | SSIM↑ | LPIPS↓ | PSNR↑ | SSIM↑ | LPIPS↓ | PSNR↑ | SSIM↑ | LPIPS↓ |
| Wu *et al.* [24] | 22.927 | 0.642 | 0.139 | 20.937 | 0.571 | 0.147 | 22.252 | 0.645 | 0.143 | 21.798 | 0.653 | 0.145 |
| Flare7K [3] | 23.939 | 0.651 | 0.138 | 22.566 | 0.652 | 0.144 | 23.386 | 0.667 | 0.141 | 23.431 | 0.648 | 0.140 |
| Flare7K++ [21] | 25.172 | 0.678 | 0.140 | 22.604 | 0.612 | 0.146 | 23.615 | 0.667 | 0.142 | 24.495 | 0.671 | 0.138 |
| FlareReal600 [34] | 23.233 | 0.617 | 0.139 | 22.587 | 0.596 | 0.147 | 24.079 | 0.658 | 0.140 | 22.821 | 0.630 | 0.157 |
| FlareX (Ours) | 25.388 | 0.682 | 0.131 | 23.882 | 0.660 | 0.138 | 25.459 | 0.692 | 0.133 | 25.096 | 0.688 | 0.131 |

## 3.4 Real-world image collection

The test images of Zhou *et al.* [32] lack ground truths, while those from Wu *et al.* [24], Flare7K [3], and FlareReal600 [34] contain almost no reflective flares. Flare7K [3] and FlareReal600 [34] simulate typical lens dirt patterns by polluting the cover glass to capture flare-corrupted images, then capture flare-free images by cleaning the front lens. However, this method fails to eliminate strong and reflective flares in the ground truths, as shown in Figure 7, which can severely distort quantitative evaluations. In some cases, cleaner flare removal by the model even receives lower scores in the residual regions due to mismatches with flawed ground truth. Besides, it is physically impossible to obtain paired images of reflective flares [22] due to the internal reflection, so only a synthetic reflective flare test set [22] is available.

To address this challenge, we propose a masking method to collect completely flare-free images. Since flares are typically caused by intense light sources, both scattering and reflective flares can be effectively removed by simply blocking these strong lights. Specifically, after capturing the flare-corrupted image, we use an eye-exam occluder to block the direct light source and then capture the flare-free image under the same conditions. We then annotate the area where the occluder is placed, and this region is excluded from the quantitative evaluation of the performance metrics. We collect 100 pairs of test images using different smartphones and professional cameras. Our comparison with existing test sets, as shown in Table 2, highlights that our benchmark better aligns with user preferences. Nearly one-third of the samples in existing test sets fail to accurately reflect the performance improvement from Flare7K [3] to Flare7K++ [21]. Additionally, off-screen flare captured by flagship smartphones with large apertures has sparked widespread online discussions. Off-screen flare appears when light enters the camera from specific angles, presenting as thick, band-like artifacts, which can not be effectively removed by existing methods. We collect 63 representative

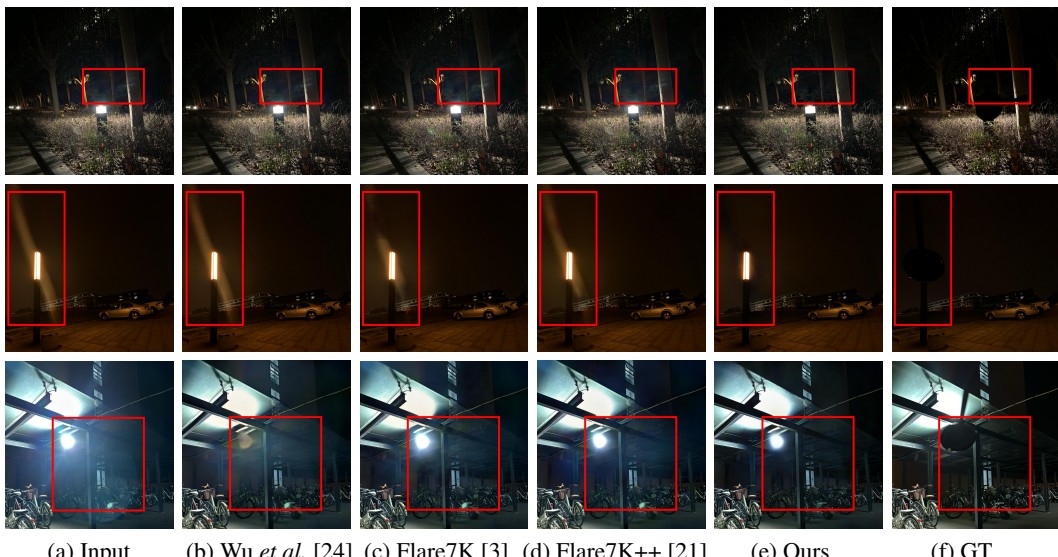

|     (a) Input     |  (b) Wu *et al.* [24]  |  (c) Flare7K [3]  |  (d) Flare7K++ [21]  |  (e) Ours  |  (f) GT  |

Figure 8: Visual comparison of flare removal on the proposed test set. The name of each column represents the dataset that is used to train Uformer. Compared to (b), (c), and (d), (e) shows the best flare removal performance.

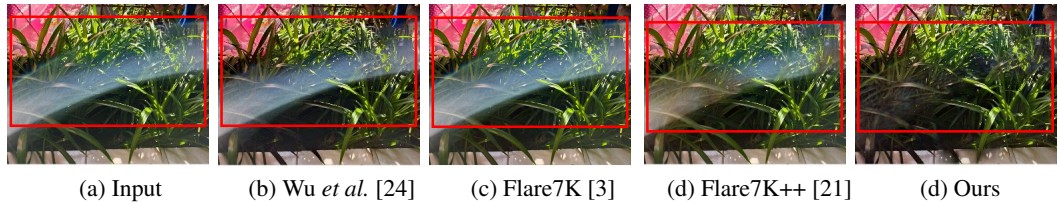

|     (a) Input     |  (b) Wu *et al.* [24]  |  (c) Flare7K [3]  |  (d) Flare7K++ [21]  |  (d) Ours  |

Figure 9: Visual comparison of off-screen flare removal. Previous methods struggle to remove these wide, white streak-like flares, whereas our approach successfully eliminates them.

images captured by various smartphones from the Internet. Our proposed masking method and collected dataset support a comprehensive evaluation of model performance in lens flare removal, particularly in assessing severe, reflective, and off-screen flare removal.

# 4 Experiments

## 4.1 Comparisons with previous datasets.

**Experiments setting.** To ensure a fair comparison, we apply the same data aggregation approach used in previous works [3, 41]. The loss functions in our work align with the previous works [3, 21, 42], comprising the $L_1$ loss, the perceptual loss with a pre-trained VGG-19 [43] and the reconstruction loss. We train the same models used in the previous benchmark [3], including HINet [37], MPRNet [38], Restormer [40], and Uformer [39], with the addition of AST [44]. We perform model training on two NVIDIA GeForce RTX 3090 GPUs with 24GB of memory. The models are trained on flare-corrupted images cropped to $512 \times 512$ resolution, with a batch size of 2, for 30,000 iterations.

**Quantitative comparison.** We adopt full-reference metrics PSNR and SSIM [45] to evaluate the performance of the models trained on different datasets. Since the flare removal is a highly perceptual task, we also use the LPIPS

Table 4: The results of off-screen flare removal.

| Metrics | Input | Wu *et al.* [24] | Flare7K [3] | Flare7K++ [21] | FlareX (Ours) |
|---|---|---|---|---|---|
| NIQE ↓ | 4.144 | 4.023 | 4.050 | 4.026 | **3.967** |
| BRISQUE ↓ | 31.894 | 29.070 | 30.621 | 29.766 | **28.354** |

distance [46]. These metrics on our proposed test set measure the flare removal performance in non-occluded regions. Due to off-screen flare images having no ground truths, we use NIQE [47] and BRISQUE [48] as no-reference assessment metrics.

Table 5: The ablation study of the laws of illumination. "✓" and "✗" indicate whether the laws of illumination are incorporated into the data synthesis pipeline.

| Dataset | Laws of illumination | PSNR↑ | SSIM↑ | LPIPS↓ |
|---|---|---|---|---|
| Flare7K [3] | ✗ | 23.386 | 0.667 | 0.141 |
| | ✓ | 23.711 | 0.675 | 0.138 |
| FlareX (Ours) | ✗ | 25.122 | 0.681 | 0.135 |
| | ✓ | 25.459 | 0.692 | 0.133 |

Table 6: The ablation study of our dataset. Training on the entire FlareX yields the best results, while training on Flare-3D alone leads to poor performance due to its limited amount.

| Method | Flare-2D | Flare-3D | PSNR↑ | SSIM↑ | LPIPS↓ |
|---|---|---|---|---|---|
| Uformer [39] | ✓ | ✗ | 24.853 | 0.690 | 0.140 |
| | ✗ | ✓ | 23.647 | 0.687 | 0.137 |
| | ✓ | ✓ | **25.459** | **0.692** | **0.133** |
| Restormer [40] | ✓ | ✗ | 24.457 | 0.616 | 0.139 |
| | ✗ | ✓ | 23.728 | 0.665 | 0.141 |
| | ✓ | ✓ | **25.096** | **0.688** | **0.131** |

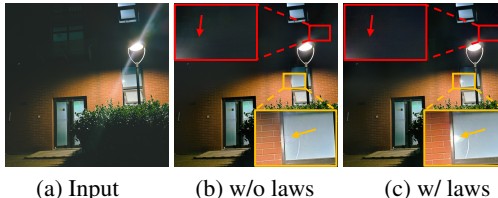

(a) Input  (b) w/o laws  (c) w/ laws

Figure 10: Visual comparison without and with the laws of illumination. The latter can effectively remove subtle flares and avoid incorrectly removing the light source in the mirror.

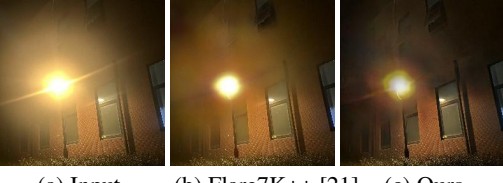

(a) Input  (b) Flare7K++ [21]  (c) Ours

Figure 11: Typical failure case of flare removal. Our method may leave noticeable artifacts when addressing images with extremely intense light sources and large flare areas.

As shown in Table 3, five baselines trained on our dataset outperform the same models trained on existing datasets. Notably, Uformer surpasses the second-best one by nearly 1.38 dB in PSNR, achieves a 3.75% increase in SSIM, and reduces LPIPS by 5%, demonstrating a significant improvement. Perceptual quality assessment results of off-screen flare removal are presented in Table 4. Uformer trained on our dataset achieves the lowest NIQE and BRISQUE scores. These results demonstrate the effectiveness of our method.

**Qualitative comparison.** To ensure fairness, we use the same Uformer trained on different datasets to conduct visual comparisons. As shown in Figure 8, the first and third rows of images illustrate that the model trained on our dataset is capable of removing reflective flare with special patterns. The third row also highlights that the model trained on the FlareX dataset excels in eliminating large-scale and intense flare patterns, outperforming previous approaches in terms of both precision and overall effectiveness. Additionally, when it comes to off-screen flare, the Uformer [39] model, trained on our comprehensive dataset, achieves superior performance compared to other methods, as illustrated in Figure 9. This indicates that our dataset plays a crucial role in enhancing the model's ability to generalize across different flare types.

### 4.2 Abalation study

**The laws of illumination.** To demonstrate the effectiveness of our synthesis method, we carry out an ablation study to assess the influence of incorporating the illumination law. We train Uformer on FlareX with the previous method and another on Flare7K [3] with our proposed method. As shown in Table 5, integrating the laws of illumination in the synthesis process can improve model performance, which also holds true for the previous Flare7K dataset [3]. Besides, we conduct a visual comparison with and without the laws of illumination. As shown in Figure 10, the laws of illumination allow the model to remove flares more accurately.

**The composition of our dataset.** In order to ascertain whether the two parts of our dataset can be combined to train an optimal model, we conduct an ablation study using part of the dataset and the whole. As demonstrated in Table 6, both Uformer and Restormer achieve optimal performance when trained on the complete dataset.

### 4.3 More visual results.

We evaluate flare removal performance on images containing multiple flares, comparing Uformer models trained on Flare7K++ and our proposed FlareX dataset (see Figure 12). As shown in the first two rows of Figure 12, the model trained on FlareX demonstrates more effective removal of dense

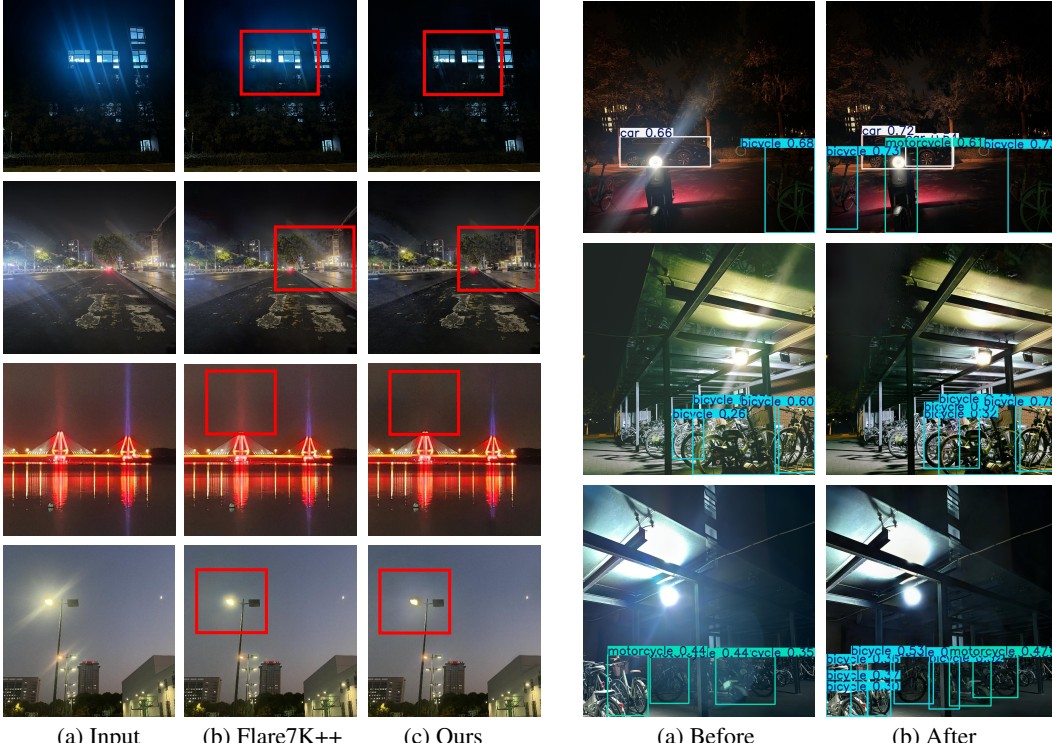

| (a) Input | (b) Flare7K++ | (c) Ours |
|-----------|---------------|----------|

| (a) Before | (b) After |
|------------|-----------|

Figure 12: Visual results of multi-flare removal. The test images come from ours (the first row), FlareReal600 [34] (the second row), and Zhou *et al.* [32] (the third and fourth row).

Figure 13: Object detection with flare-corrupted and flare-removed images. Flares can obscure objects such as bicycles and motorcycles, making them undetectable.

flares. Furthermore, in the fourth row, which presents a scenario involving both streak and glare flares, the FlareX-trained model also achieves cleaner and more precise removal.

We perform a visual comparison of object detection before and after flare removal. As illustrated in Figure 13, in the first and second rows, the flare-removed images reveal previously occluded objects, such as bicycles and motorcycles, which are not detected in the flare-corrupted images. Besides, in the third row, flares cause the model to mistakenly classify bicycles as motorcycles, whereas this issue is resolved after flare removal. This experiment indicates that lens flare can negatively impact the model, thereby presenting a significant threat to high-level applications.

## 5 Conclusion

In this paper, we present FlareX, a physics-informed dataset designed to advance the task of lens flare removal. First, to enhance pattern diversity, we generate 9,500 flare templates based on 95 physically flare types using a Blender plugin. Second, to improve the realism of synthetic data, we incorporate the laws of illumination into the 2D flare synthesis process, addressing the issue of unnatural brightness distributions common in existing methods. Finally, to further bridge the gap between synthetic and real-world images, we construct 60 scenes and render 3,000 paired samples using a physically-based rendering pipeline. Additionally, to enable more reliable real-world evaluation, we propose a masking strategy to collect 100 pairs of flare and flare-free images, effectively avoiding residual flare artifacts introduced by lens-wiping methods. Extensive experiments and ablation studies demonstrate the effectiveness of our dataset in improving flare removal performance and generalization across various models, paving the way for future research in this field.

**Limitations.** Despite the effectiveness of current models, extremely heavy flare remains a significant challenge, often resulting in visible artifacts after restoration, as shown in Figure 11. Future research may benefit from incorporating physical priors or transferring structural cues from cleaner regions to better recover severely degraded content.

**Acknowledgement:** This work was supported by Shenzhen Science and Technology Program (No. JCYJ20240813114229039), National Natural Science Foundation of China (Nos. 624B2072, U22B2049, 62302240), Natural Science Foundation of Tianjin (No.24JCZXJC00040), the Funding from SCCI, Dalian Univ. of Technology (No. SCCI2023YB01), and Supercomputing Center of Nankai University.

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
