# OpenReview forum: "FlareX: A Physics-Informed Dataset for Lens Flare Removal via 2D Synthesis and 3D Rendering"
_NeurIPS.cc/2025/Datasets_and_Benchmarks_Track — NeurIPS 2025 Datasets and Benchmarks Track poster_

### Official Review · Reviewer_5ETG · 2025-06-06

**Rating:** 4
**Confidence:** 5

**Summary:**

In this submissions, the authors propose FlareX, a dataset targeting image lens flare removal, proving particularly helpful in reflective flare representation at the data level.
The data collection provides a large collection of flare patterns, simulated using Blender's Flared2  plugin. The patterns belong to both 2D and 3D setups, capturing lens flare behavior for different camera poses relative to the position of the illuminators.
On top of the simulated data, the authors propose a collection of real samples, for which the reference images are acquired using illuminator occlusion and masking. Annotations for illuminators, flare types, and  occlusion masks are also provided.
A number of standard image restoration methods are validated on FlareX.

**Additional Feedback:**

Even if the field is in need of development, lacks data contributions, and the visual results are clearly positive, a submission for flare removal in which none of the flare-driving factors are discussed or documented needs a massive overhaul. A lens flare removal without any connection to representative optical systems and their properties, what is the parameter range represented at the data level, and what are the applications that can be targeted through the represented range, can not gain enough confidence for a positive recommendation.

**Dataset Code Accessibility:**

Yes

**Dataset Code Comments:**

A BasicSR compatible dataloader is provided in the code submitted as supplementary material.

**Ethical Considerations:**

No, there are no or only very minor ethics concerns

**Final Justification:**

On the positive side, the addition of newer models in the field of lens flare removal increases the quality of the manuscript. Also, the statistics regarding masked area size in testing clear some of the doubts.
On the negative side, the authors chose not to provide any details regarding the parameters used in simulation, the available lens or aperture models. It can be acknowledged that the simulated data correlates with a benchmark composed of real data, but this is already explored in Flare7K++. In terms of contributions, the dataset is likely advantageous in quantitative terms, but the quality degree is likely similar to previous datasets. Adding discussions showcasing the advantages in simulation vs. using AfterEffects (Flare7K/Flare7K++), or Adobe Photoshop [SFNet, ICIP24], [MIPI24 Image Flare Removal Challenge],   would help showcasing the advantages of the submitted work. Therefore, even though the general opinion is of a borderline work, I tend towards a positive decision.

**Limitations Weaknesses:**

1. None of the parameters describing flare appearance and behavior are discussed in the main manuscript. Thus, there is a number of questions needing an answer:
      a) What are the properties of the optical systems modeled in simulation, and how do they compare to the optical systems representative today for different photography applications? The number of elements, number of groups, group placement, coating technology, moving groups effects have to be taken into account.
     b) How do the optical systems used in simulation relate to the smartphone photography segment, which is widely addressed in the submitted paper?
     c) What is the connection between the parameters available for simulation in Flared2 and the properties of the optical systems relevant for questions a, b? What are the advantages of using Flared2 compared to collections from Adobe AfterEffects or Photoshop?
     d) Camera aperture has a significant effect in flare appearance. Thus, aperture types have to be discussed, the number of blades, if the aperture types available for simulation are aligned with current aperture mechanisms implemented in current DSLR, DSLM cameras, or smartphone cameras.
2. The optical systems of the devices used in data collection are not described in the submitted materials.
3. The real-data acquisition is based on light source occlusion. However, when the light source is larger, the occluder has to be placed closer to the camera, covering a large image segment. As the occluded area has to be excluded from the evaluation, what does an evaluation on such an image pair say about the model performance? Statistics regarding the ratio of the pixels excluded from the evaluation could be helpful in assessing the quality of the data.
4. In recent years, the field received significant attention through public benchmarks and competition, driving development of new models and algorithms. However, the benchmark provides evaluation for image restoration models proposed in 2021 and 2022. Newer methods, especially those proposed in the context of lens-flare removal, have to be considered.

**Strengths Contributions:**

1. The manuscript is well written, and quite easy to understand.
2. The data collection seems to be more aligned with Flare7K++, which provides partly real samples characteristic to smartphone cameras and a APS-C kit lens. Even if Flare7K++ proved its limits in terms of reference image flare appearance, it remains a good work in the field.

---

> ### Author Rebuttal · Authors · 2025-07-29
>
> Thank you very much for your professional and valuable comments. We have addressed your concerns as follows.
>
> >*a) What are the properties of the optical systems modeled in simulation, and how do they compare to the optical systems representative today for different photography applications? The number of elements, number of groups, group placement, coating technology, moving groups effects have to be taken into account.*
>
> Our dataset design principle bears great resemblance to the previous "Flare7K: A Phenomenological Nighttime Flare Removal Dataset".
> Their original explanation about the phenomenological design is stated as follows: "In a real-world situation, the diversity of aperture dirt and lens system makes it really difficult to design an aperture function that is same as the type of target lens flare. Thus, we design a phenomenological method to synthesize lens flares rather than using a physics-based flare generation method. Our phenomenological method still produces reasonable flares."
> Therefore, we also do not directly design optical systems. Instead, we primarily simulate flares by varying the number of reflections, creating diverse shapes, adjusting focal lengths, and modifying light conditions (e.g., color and intensity).
> Experimental results demonstrate that this data is also meaningful for the neural network.
>
> > *b) How do the optical systems used in simulation relate to the smartphone photography segment, which is widely addressed in the submitted paper?*
>
> Smartphone lens groups have fewer elements compared to professional cameras, and are more prone to daily stains (such as grease and dust) than camera lenses, resulting in more severe flare phenomena than professional lenses. Therefore, during the dataset creation process, we simulated many of these common flare artifacts that occur in smartphones.
>
> > *c) What is the connection between the parameters available for simulation in Flared2 and the properties of the optical systems relevant for questions a, b? What are the advantages of using Flared2 compared to collections from Adobe AfterEffects or Photoshop?*
>
> The parameters in Flared are indirectly related to real optical systems. For example, the number and shape of streaks correspond to the type of aperture; the number of reflections in the flare corresponds to the number of lens groups; and the intensity of glare corresponds to the severity of aperture dirt or variations in coating techniques. Blender has more physical constraints and manipulable properties than Adobe After Effects and Photoshop in 2D.
>
> > *d) Camera aperture has a significant effect in flare appearance. Thus, aperture types have to be discussed, the number of blades, if the aperture types available for simulation are aligned with current aperture mechanisms implemented in current DSLR, DSLM cameras, or smartphone cameras.*
>
> In our simulation, we considered whether we should fix the shape of the scattering flare to match the aperture of the equipment we used for shooting. This approach may yield better results on our test set, but it narrowed the scope of the dataset and may not be well-suited for adapting to various other types of equipment.
> Therefore, in our simulation, we opted for a wider range of flare types to accommodate a variety of devices.
>
> Taking into account all your valuable suggestions, we will incorporate the discussions on the flare production process into the main text. Thank you once again for your professional and patient advice.
>
> >*2. The optical systems of the devices used in data collection are not described in the submitted materials.The real-data acquisition is based on light source occlusion. However, when the light source is larger, the occluder has to be placed closer to the camera, covering a large image segment. As the occluded area has to be excluded from the evaluation, what does an evaluation on such an image pair say about the model performance? Statistics regarding the ratio of the pixels excluded from the evaluation could be helpful in assessing the quality of the data.*
>
> The test set we collected comes from an iPhone 14 Pro, a Xiaomi 14, and a Canon EOS R7 with a 15–150mm f/2.8 lens.
>
> The presence of large residual flares in the ground truth of previous datasets may yield results that are contrary to user perception. For example, if an area of the image has been repaired cleanly, but due to the presence of flares in the GT, it may receive a lower score. Conversely, if the restored image is not clean, it may still receive a higher score because it is close to the unclean GT. Using our test set may reduce the performance gap between the two results, but it will not produce opposite results. As shown in Table 2 of the manuscript, our test metrics are more aligned with user evaluations.
> Thank you for your suggestion on calculating the occluded area. The average ratio of the masked area is 5.08%, which has a minimal impact on the evaluation. The result can indeed further demonstrate the persuasiveness of the test set.
>
>
> >*3. In recent years, the field received significant attention through public benchmarks and competition, driving development of new models and algorithms. However, the benchmark provides evaluation for image restoration models proposed in 2021 and 2022. Newer methods, especially those proposed in the context of lens-flare removal, have to be considered.*
>
> Thank you for your constructive comments. As suggested, we have included more recent deep learning-based methods, particularly those related to flare removal.
>
> | Dataset      |   |            | FF-former [1] |           |   |            | Difflare [2] |           |   |            |  AST [3]  |           |   |            | Flare-Free Vision [4] |           |
> |--------------|:-:|:----------:|:-------------:|:---------:|:-:|:----------:|:------------:|:---------:|:-:|:----------:|:---------:|:---------:|:-:|:----------:|:---------------------:|:---------:|
> |              |   |    PSNR    |      SSIM     |   LPIPS   |   |    PSNR    |     SSIM     |   LPIPS   |   |    PSNR    |    SSIM   |   LPIPS   |   |    PSNR    |          SSIM         |   LPIPS   |
> | Wu et al.    |   |   24.956   |     0.628     |   0.147   |   |   24.522   |     0.629    |   0.140   |   |   24.845   |   0.614   |   0.142   |   |   24.653   |         0.625         |   0.142   |
> | Flare7K      |   |   24.882   |     0.609     |   0.140   |   |   24.834   |     0.638    |   0.137   |   |   24.261   |   0.623   |   0.143   |   |   24.290   |         0.614         |   0.139   |
> | Flare7K++    |   |   24.134   |     0.634     |   0.142   |   |   24.975   |     0.652    |   0.139   |   |   24.820   |   0.642   |   0.140   |   |   24.425   |         0.638         |   0.141   |
> | FlareReal600 |   |   24.901   |     0.592     |   0.159   |   |   25.096   |     0.614    |   0.138   |   |   24.550   |   0.623   |   0.139   |   |   24.891   |         0.602         |   0.141   |
> | Ours         |   | **25.377** |   **0.650**   | **0.134** |   | **25.219** |   **0.666**  | **0.132** |   | **25.379** | **0.651** | **0.135** |   | **25.478** |       **0.659**       | **0.133** |
>
> [1] Dafeng Zhang, Ouyang Jia, Guangun Liu, Xiaobing Wang, Xiangyu Kong and Zhezhu Jin. Ff-former: Swin fourier transformer for nighttime flare removal. In CVPRW 2023.
>
> [2] Tianwen Zhou, Qihao Duan and Zitong Yu. Difflare: Removing image lens flare with latent diffusion model. In BMVC, 2024.
>
> [3] Shihao Zhou, Duosheng  Chen, Jinshan Pan, Jinglei Shi and Jufeng Yang. Adapt or Perish: Adaptive Sparse Transformer with Attentive Feature Refinement for Image Restoration. In CVPR 2024.
>
> [4] Kotp, Yousef and Torki, Marwan. Flare-free vision: Empowering uformer with depth insights. In ICASSP 2024.

---

> ### Comment · Reviewer_5ETG · 2025-08-04
> **Final rating discussion**
>
> Dear authors,
>
> Thanks for your rebuttal. The information you provided answers the questions regarding the quality of the test data, given the size and the position of the flare occluder. Also, the additional comparisons including newer models, especially those tailored for flare removal in particular, increase the quality of your submission.
> The main weakness remains the documentation of the simulation parameters. Typically, the existing simulators focus on rather old lens models (given the scarcity of specs for newer designs). The opinion is that, even if it is impossible to study in detail a wide number of devices, a study regarding how a limited set of parameters from simulation aligns with the properties of devices representative for some photography applications would greatly benefit the quality of the proposed paper.
> As it is, the data contribution represents an iteration over public datasets like Flare7K++, with the simulation replaced by a software with arguably improved functions. The opinion is that showcasing these advantages is crucial for a positive feedback.
> In terms of real data, the main issue of Flare7K++ is the quality of the reference images (with many affected by flares to a limited degree). Even if the occluder strategy can improve in this aspect, it is not difficult to find samples (for example in the test_dataset, reference samples 3, 4, 5,...), which suffer from the same problems (the effect of the meteo conditions in this manifestation is impossible to assess).
> To conclude, the additional information provided in the rebuttal clears some doubts, and some additional comparisons benefit the submission, but some questions regarding the incremental quality refinement w.r.t. existing literature still remain.

---

> > ### Author Response · Authors · 2025-08-04
> > **Further explanation on the simulation parameters**
> >
> > Thank you for your additional valuable feedback.
> >
> > To explicitly map Flared2 parameters to physical camera lens properties, we gather some common lens parameters and captured images from the Internet. These parameters related to flares include focal length, number of elements, number of aperture blades, and coating.
> >
> > | Target Device      | Key Flare Phenomenon       | Flared2 Control Parameter     | Physical Basis                                  |
> > |--------------------|----------------------------|-------------------------------|-------------------------------------------------|
> > | Nikon 50mm f/1.8G  | Circular ghost reflections|  Reflection Setting: Count=8  | 9-element lens design → 8 internal reflections  |
> > | Nikon 50mm f/1.8D  | Hexagonal ghosts |  Ghosts Generator: Shape=Hexagon, Count= 6  | 6 aperture blades → 6 hexagonal ghosts |
> > | M.ZUIKO DIGITAL ED 40-150mm F4.0 PRO         | 14-point sunstars          | Starburst Points = 14          | 7 aperture blades → 14 points (2× blades)       |
> > | Sirui Anamorphic   | Horizontal streaks         | Angle of the streak = 0 and Anamorphic Factor = 1.33x   | Cylindrical lens element compression ratio     |
> > | Sony 24-70mm GM II | Subtle veiling glare    | Glare Intensity = 0.3 | Nano AR coating reduces scatter → weaker glow  |
> > | iPhone 15 Pro Main Camera (IMX803) | Rainbow-colored halos     | Halo Generator: Yes and Focus Length Setting = 22/48 mm   | Multi-layer coating diffraction on small lenses and Focus length = 22/48 mm|
> >
> > As shown in the table, Flared2 can simulate different reflection times according to different numbers of lens elements. The shape of the ghosts and the number of streaks in the starburst are adjusted based on the number of aperture blades. The Sirui Anamorphic lens used in film shooting often produces horizontal streaks, which can also be simulated by Flared2. The intensity of glare can be controlled depending on different coating techniques. It can also simulate halos and hoops under strong light in iPhone.
> >
> > It should be noted that prior methods can not dynamically simulate these properties. The primary focus of Flare7K is to simulate **the aperture dirt function** that exists on the surface of all lenses, including glare, streaks, and shimmer (as stated on page 3 of their supplementary material). Therefore, Flare7K is not designed to model lens-specific flare patterns for individual cameras, especially reflective flares.
> >
> > Admittedly, varying lens designs produce diverse flares that cannot be precisely simulated in the current flare removal research. However, leveraging more comprehensive parametric controls, we have created a wide array of lens flare templates that significantly benefit data-driven training. We can also manually adjust shapes, colors, and counts of flare to match the optical characteristics of specific cameras based on their captured images. Furthermore, constructing flares within Blender’s 3D space enables superior spatial positioning modeling, as shown in the demo of supplementary materials.
> >
> > Regarding the paired image capture method, our occlusion-based test set enables a more accurate assessment of user preference (Table 2 of the manuscript). Unlike the previous test set, which contained large areas of flare, the test sample you mentioned only includes a very small flare region. After add this area into the mask, we found that it has a negligible impact (<0.01 dB in PSNR) on the quantitative results. We will refine the imperfect test samples you identified in the revisions. Theoretically, this occlusion approach can achieve completely flare-free images as we discussed before.
> >
> > Thank you very much for the detailed discussion, which helped us better clarify the positioning of our work. Previous studies mainly focused on constructing general flare patterns caused by variations in the dirt function on the **lens surface**. In contrast, our work goes a step further by simulating flare patterns resulting from differences in the internal structure of the lens.

---

> ### Author Response · Authors · 2025-08-06
> **Please let us know if you have any concerns. Eager to hearing from you!**
>
> Dear Reviewer 5ETG,
>
> Thank you very much for your effort and professional comments throughout the review process. We hope our previous responses have addressed your concerns. If you have any further feedback or a final decision, please let us know.
> We look forward to hearing from you and continuing to discuss the work.
>
> Thank you again for your time and guidance!
>
> Best wishes,
>
> Authors of Paper 1188

---

> > ### Comment · Reviewer_5ETG · 2025-08-06
> > **Message to authors**
> >
> > Thank you for the additional clarifications. The arguments represent indeed advantages of the proposed work.
> > Based on the prospects of improving the presentation regarding the advantages in simulation, compared to the previously proposed datasets, a score increase is possible.

---

> > > ### Author Response · Authors · 2025-08-06
> > > **Thanks for Guidence and Support!**
> > >
> > > Dear Reviewer 5ETG,
> > >
> > > Thank you very much for your time and expert guidance throughout the rebuttal process. We will incorporate your valuable suggestions into the paper, and hope this will help advance future research on flare removal.
> > >
> > > Sincerely,
> > >
> > > The Authors of Paper 1188

---

### Official Review · Reviewer_mFc9 · 2025-06-22

**Rating:** 4
**Confidence:** 4

**Summary:**

This paper proposes a 2D and 3D flare simulation method using the FLAERD plugin in Blender. Based on illuminance laws, a 2D flare image synthesis method was developed to enhance the realism of flare images. Furthermore, physical consistency is further enhanced through 3D rendering techniques.

**Dataset Code Accessibility:**

Yes

**Ethical Considerations:**

No, there are no or only very minor ethics concerns

**Final Justification:**

The FlareX is richer than Flare7k in terms of flare samples and can be used as an extension of Flare7k to improve the robustness of flare removal in real scenes.

**Limitations Weaknesses:**

1. For indoor scenes with shallow depth in background images, depth estimation is relatively accurate. However, for outdoor scenes with deeper depth, does this method remain effective? How much performance improvement is there compared to random fusion?
2. The restoration effect at the light source is challenging and significantly impacts objective metrics. However, the FlareX test set masks the light sources, so adding test results with light sources, including objective and visual results, would be beneficial.
3. Objective metrics for FlareReal600 are reported in Table 3, but the corresponding visualization results are missing from Figure 7.
4. Both Flare7k's real and FlareReal600 datasets consist of real-world test scenes, and testing on these datasets better demonstrates FlareX's robustness. While unavoidable small reflective flares exist in both, their overall impact is minimal. However, based on the results in the appendix, the performance of the Uformer trained on FlareX in these two test sets is poor, raising concerns about the dataset's robustness in real-world scenarios.
5. Table 6 does not prove the effectiveness of Flare-3D, it merely indicates that incorporating multiple templates improves model performance. To better validate Flare-3D's efficacy, conducting ablation experiments by rendering corresponding background images and Flare images and fusing them in 2D would be more convincing.
6. Within the provided 2D flare dataset, each light source exhibits a notably wide and prominent band-shaped artifact. What is the underlying cause of this phenomenon, and does it introduce potential impacts on the performance?
7. I used the Flare-2D and Flare-3D datasets to train the Uformer, and regardless of whether the model was trained individually or in combination, I observed that the performance was consistently unsatisfactory on the Flare7K real test dataset. This raises a key question about the robustness of the dataset in real-world scenarios.

**Strengths Contributions:**

1. This paper is well-organized, and the proposed methods have been experimentally validated.
2. The proposed dataset exhibits higher realism and has significant practical value.

---

> ### Author Rebuttal · Authors · 2025-07-29
>
> We appreciate the positive and constructive comments on our paper. The raised concerns are addressed as follows.
>
> >*For indoor scenes with shallow depth in background images, depth estimation is relatively accurate. However, for outdoor scenes with deeper depth, does this method remain effective? How much performance improvement is there compared to random fusion?*
>
> When the monocular depth estimation is relatively accurate, our method will yield better results. Detailed ablation experiment results can be found in Table 5 and Figure 9 of the main text.
> If the depth estimation is totally incorrect, the synthesis method's performance just degrades to the same level as random synthesis of previous work.
>
> >*The restoration effect at the light source is challenging and significantly impacts objective metrics. However, the FlareX test set masks the light sources, so adding test results with light sources, including objective and visual results, would be beneficial.*
>
> The reason we block the light source is to obtain a completely clean area for GTs (the wiping method will leave behind a large area of reflection and residual flares).
> When we conduct the user study (Table 2 in the manuscript), we find that the users' preferences were more consistent with the quantitative evaluation results of our test set. This suggests that for humans, achieving a clean large-scale flare area is more critical than addressing flare regions near the light source.
>
> Although it would be ideal to obtain ground truth images in which all regions are flare-free, achieving this remains highly challenging at the current stage and will be an important direction for future research.
> Therefore, a practical solution is to conduct evaluations by combining our dataset with previous flare-related datasets that include light sources.
> We conduct comparative experiments on all other previous test sets with light sources (please refer to the supplementary materials for objective results in Table 3 and visual results in Figures 6 and 8).
>
> >*Objective metrics for FlareReal600 are reported in Table 3, but the corresponding visualization results are missing from Figure 7.*
>
> Thanks for the suggestion! We will incorporate it into Figure 7 of the subsequent version.
>
> >*Both Flare7k's real and FlareReal600 datasets consist of real-world test scenes, and testing on these datasets better demonstrates FlareX's robustness. While unavoidable small reflective flares exist in both, their overall impact is minimal. However, based on the results in the appendix, the performance of the Uformer trained on FlareX in these two test sets is poor, raising concerns about the dataset's robustness in real-world scenarios.*
>
> In fact, each training set performs best on its own test set, due to the domain gap between different datasets (Table 3 in the supplementary materials). For instance, Flare7K++ performs best on its own test set, but performs the worst on the test set used by Wu et al. Similarly, FlareReal600 performs best on its own test set, but performs worse than Ours on the other test sets.
> Our results exhibited relatively stable performance (first or second place) across all test sets. The final average PSNR value across all test sets was 1.127 dB higher than that of the second-place model, which demonstrates the robustness of our dataset.
>
> >*Table 6 does not prove the effectiveness of Flare-3D, it merely indicates that incorporating multiple templates improves model performance. To better validate Flare-3D's efficacy, conducting ablation experiments by rendering corresponding background images and Flare images and fusing them in 2D would be more convincing.*
>
> Following the valuable suggestion, ablation studies with separate rendering are presented below.
> The results show that rather than simply adding more templates, rendering flare images with stronger physical constraints in 3D leads to more significant performance gains.
>
> |                              |   |  PSNR $\uparrow$ |  SSIM $\uparrow$| LPIPS $\downarrow$|
> |------------------------------|---|:------:|:-----:|:-----:|
> | Flare-2D                     |   | 25.421 | 0.641 | 0.140 |
> | Flare-2D + Flare-3D (synthesized in 2D) |   | 25.480 | 0.644 | 0.139 |
> | Flare-2D + Flare-3D (rendered in 3D)    |   | **25.688** | **0.658** | **0.133** |
>
> >*Within the provided 2D flare dataset, each light source exhibits a notably wide and prominent band-shaped artifact. What is the underlying cause of this phenomenon, and does it introduce potential impacts on the performance?*
>
> This band-shaped pattern was specifically introduced to simulate severe smearing-type lens flare degradation, and it was only present in the first 400 images of the dataset. To verify its impact on test results, we excluded these first 400 images. To control for the effect of reduced data volume, we also excluded the last 400 images for comparison. Experimental results show that this flare pattern is beneficial for flare removal; its removal actually degrades performance as the similar as other images.
>
> |                              |   |  PSNR $\uparrow$ |  SSIM $\uparrow$| LPIPS $\downarrow$|
> |------------------------------|---|:------:|:-----:|:-----:|
> | without these first 400 images|   | 25.582 | 0.654 | 0.137 |
> | without other 400 images |   | 25.579 | 0.655 | 0.138 |
> | Full                     |   | **25.688** | **0.658** | **0.133** |
>
> >*I used the Flare-2D and Flare-3D datasets to train the Uformer, and regardless of whether the model was trained individually or in combination, I observed that the performance was consistently unsatisfactory on the Flare7K real test dataset. This raises a key question about the robustness of the dataset in real-world scenarios.*
>
> Thank you very much for taking the time to follow our work. There may be some differences in data augmentation and training parameter settings. We will release detailed synthesis, training details, and pre-trained models on GitHub after camera-ready.

---

> > ### Comment · Reviewer_mFc9 · 2025-08-04
> >
> > I have another question, when I tested your test set, the A6000 GPU (48G) was not enough to complete the inference, so what was the resolution when you were doing the inference? How much GPU memory is required to perform inference at the original image size (3024x3024)?

---

> > > ### Author Response · Authors · 2025-08-04
> > > **The GPU memory for testing**
> > >
> > > Dear Reviewer,
> > >
> > > Thank you very much for your response. The GPU memory usage during testing depends on the model being used. If using Uformer, direct inference on 3024×3024 images is not feasible in 48GB. For high-resolution inference, we adopted the strategy provided in the FlareReal600 GitHub repository. The core idea is to downscale the input image for inference, then compute the residual between the inference result and the resized image, and finally add this residual back into the original high-resolution image.
> > >
> > > ```
> > > deflare_img = merge_img_org.cuda().unsqueeze(0) - resize2org(merge_img - deflare_img)
> > > ```
> > >
> > > Sincerely,
> > >
> > > Authors of paper 1188

---

> > > ### Comment · Area_Chair_KgFD · 2025-08-06
> > > **Please read the author’s responses**
> > >
> > > Please read the author’s responses, update review to include final justification, update rating and submit mandatory acknowledgement for author response. Thanks!

---

### Official Review · Reviewer_arqH · 2025-06-23

**Rating:** 5
**Confidence:** 5

**Summary:**

For mobile phone cameras, flare is a significant cause of poor imaging quality. How to efficiently and accurately remove flares is a very challenging task, and its main difficulty lies in the difficulty of obtaining high-quality training data pairs (clean images and flare images). The traditional approach is to manually add flares on clean images, but the diversity of flares added in this way is insufficient and sometimes violates physical laws. To address these challenges, this work proposes new stratergies to build physics-informed dataset with both 2D synthesis and 3D rendering. The paper is well written, with solid idea and sufficient and persuasive experiments.

**Dataset Code Accessibility:**

Yes

**Ethical Considerations:**

No, there are no or only very minor ethics concerns

**Final Justification:**

The authors have well addressed my concerns, and I would like to maintain my positive rating.

**Limitations Weaknesses:**

In the section of experiment, more discussion of the results and insights are suggested to be added into the manuscript. For example, when Table 4 and Figure 8 show the superiority of the proposed FlareX, could the authors provide more detailed analysis on the improvment?

**Strengths Contributions:**

1. The authors build a significant dataset FlareX, which offers almost 10K 2D templates derived from 95 flare patterns and 3K flare image pairs rendered from 60 3D scenes. They also design a masking approach to obtain real-world flare-free images from the corrupted counterparts to measure the performance of the model on real-world images. The reviewer believes this new dataset will attract more researchers' attention to the study of flares removal, and then promote the rapid development of this important field.
2. The authors carry out extensive experiments and ablation studies, which prove the effectiveness of FlareX in improving flare removal performance and generalization across various models.
3. The writing of the paper is smooth, with exquisite and clear illustrations. The proposed dataset construction scheme is well explained and easy to understand.
4. The demo attached in the supplementary material is fantasic.

---

> ### Author Rebuttal · Authors · 2025-07-29
>
> Thank you very much for your high recognition of our work and your valuable suggestions. Our response is as follows:
>
> >*In the section of experiment, more discussion of the results and insights are suggested to be added into the manuscript. For example, when Table 4 and Figure 8 show the superiority of the proposed FlareX, could the authors provide more detailed analysis on the improvment?*
>
> We will incorporate these representative supplementary materials and discussions into the main text in the subsequent version.
> In Table 4, the improvement in the streak region is often greater than that in the glare region. This can be attributed to the smaller influence range of streaks, allowing the model to learn similar information from nearby clean regions. However, this is difficult for the large-scale glare.
> Regarding the restoration effect of multiple light sources in Figure 8, since our 2D dataset is based on deep synthesis and our 3D dataset is based on spatial rendering, the model's perception of light sources will be more accurate, especially for multiple light sources at different positions, resulting in better restoration effects.

---

> > ### Comment · Reviewer_arqH · 2025-08-05
> >
> > I appreciate the authors’ responses. They have adequately addressed my concern, and I will retain my positive score.

---

> > > ### Author Response · Authors · 2025-08-05
> > > **Thanks for Recognition!**
> > >
> > > Dear Reviewer arqH,
> > >
> > > We sincerely appreciate your feedback and support. We will incorporate the suggestions you mentioned into the manuscript. Thank you again for your recognition of our work.
> > >
> > > Best wishes,
> > > Authors of paper 1188

---

### Official Review · Reviewer_amm4 · 2025-07-03

**Rating:** 4
**Confidence:** 4

**Summary:**

The paper contributes by showing a large physics dataset that combines 2D synthesis with 3D rendering. The authors generated 9,500 flare templates, a brightness adjustment process based on the laws of illumination, and created 3,000 paired 3D renderings in Blender. In addition, they introduce masking techniques and collect 100 pairs of real flare/no-flare images, ensuring that the dataset not only scales well, but also provides reliable ground truth for evaluation. Experimental results show that training on FlareX results in consistent performance gains.

**Dataset Code Accessibility:**

Yes

**Ethical Considerations:**

No, there are no or only very minor ethics concerns

**Final Justification:**

The authors' responses have addressed my concerns. This paper provides a comprehensive dataset for 2D and 3D rendering. Although my decision is on the borderline, I tend to accept this paper.

**Limitations Weaknesses:**

Monocular Depth Dependence: The 2D compositing stage relies heavily on monocular depth estimation, and if the depth prediction is wrong, it will directly affect the accuracy of the luminance correction, which in turn will lead to deviations in the physical consistency of the synthesized image. Is it possible to incorporate stereo or multi-view information to improve depth reliability?
Limitations of the real test set: The paper only provides 100 pairs of real scene images, mostly outdoor single light source, indoor low light, complex multiple light sources or dynamic scenes lack of evaluation. Is this scale and diversity of testing sufficient to validate the reliability of the method in complex real-world environments?
Real-time and deployment: although the size of the dataset is sizable, the synthesis and training costs are high. Facing the demand of real-time mobile or in-vehicle system, how to ensure the de-glare effect while taking into account the model lightweight and inference speed?

**Strengths Contributions:**

The paper is generally well organized and easy to understand. However, some sections contain dense multiple statements. Some of the transitions between formula derivations and narrative descriptions also seem a bit abrupt.
Unlike approaches that rely solely on image compositing or full scene rendering, FlareX introduces the Law of Illumination in the 2D compositing phase and Blender ray tracing in the 3D phase to achieve a highly realistic scene. This design ensures the efficiency of large-scale data generation and enhances the physical realism of the data.

---

> ### Author Rebuttal · Authors · 2025-07-29
>
> We sincerely thank you for reviewing our paper and providing us with valuable feedback. We have addressed your concerns as follows.
> > *Monocular Depth Dependence: The 2D compositing stage relies heavily on monocular depth estimation, and if the depth prediction is wrong, it will directly affect the accuracy of the luminance correction, which in turn will lead to deviations in the physical consistency of the synthesized image. Is it possible to incorporate stereo or multi-view information to improve depth reliability?*
>
> Thank you for your constructive comments. When the monocular depth estimation is relatively accurate, our method will yield better results. Although there are a few cases where the depth estimation is incorrect, our synthesis method will just degrade to the same level as random synthesis. Therefore, using depth estimation yields better average results than omitting it, as evidenced by the data in Table 5.
>
> Our method is a demonstration on a large-scale, unlabeled dataset. If there is a large-scale dataset with accurate multi-view depth estimation annotations, our method is equally applicable, as we only require the depth distribution information of the images.
>
> > *Limitations of the real test set: The paper only provides 100 pairs of real scene images, mostly outdoor single light source, indoor low light, complex multiple light sources or dynamic scenes lack of evaluation. Is this scale and diversity of testing sufficient to validate the reliability of the method in complex real-world environments?*
>
> We appreciate the suggestion regarding the scale of the test set. In addition to our test set, the currently major flare test sets include the following: Wu et al. (20 images), Flare7K and Flare7K++ (shared test set of 100 images), and FlareReal600 (50 images). A comparison with these test sets is provided in the supplementary material (Table 2 and Table 3).
> We analyzed the diversity of existing test sets, as shown below. It can be observed that the distribution in existing datasets is unbalanced. In contrast, our test set achieves a more reasonable proportion across multiple categories. Besides, the greatest advantage of our test set lies in the complete removal of flares in GTs.
>
> At the current stage, in addition to our test set, we have also conducted a comprehensive evaluation by combining all test sets, as shown in Table 3 of the supplementary material.
> In future work, we will expand our test dataset to include a wider variety of scenes.
>
> |              | Outdoor : indoor | Night : Day | Single : Multiple light|
> |--------------|:---------------:|:---------:|:----------------------:|
> | Wu et al.    |       16 : 4      |    0 : 20   |          20 : 0       |
> | Flare7K      |      73 : 27     |   100: 0  |         71 : 29          |
> | FlareReal600 |      49 : 1     |    50 : 0   |          15 : 35         |
> | Ours         |      72 : 28     |   71 : 29  |          77 : 23         |
>
> >*Real-time and deployment: although the size of the dataset is sizable, the synthesis and training costs are high. Facing the demand of real-time mobile or in-vehicle system, how to ensure the de-glare effect while taking into account the model lightweight and inference speed?*
>
> Thank you for your valuable feedback. Similar to the previous Flare7K, we have utilized third-party tools, and the costs for synthesis and training are also comparable.
> Regarding the lightweight and inference speed of the model, this issue is not the focus of our research in this paper (the purpose of this paper is to obtain a higher-quality flare dataset). The lightweighting of the model will become the research focus in future work. For example, we may first detect flare regions and then apply processing only to those regions. Additionally, model distillation can be utilized to reduce computational complexity, thereby facilitating efficient deployment on edge devices.

---

> > ### Comment · Reviewer_amm4 · 2025-08-06
> > **Thanks for authors' responses.**
> >
> > The authors' responses have addressed my concerns. This paper provides a comprehensive dataset for 2D and 3D rendering. Although my decision is on the boarderline, I tend to accept this paper.

---

> > > ### Author Response · Authors · 2025-08-06
> > > **Thanks for Support!**
> > >
> > > Dear Reviewer amm4,
> > >
> > > Thank you very much for your valuable time and effort throughout the review process. We will carefully incorporate your suggestions into the revised manuscript.
> > >
> > > Best regards,
> > >
> > > The Authors of Paper 1188

---

> ### Comment · Area_Chair_KgFD · 2025-08-06
> **Please read the author’s responses**
>
> Please read the author’s responses, update review to include final justification, update rating and submit mandatory acknowledgement for author response. Thanks!

---

### Note · Authors · 2025-08-12

Dear Area Chairs and Reviewers,

We are deeply grateful for the time, effort, and constructive feedback you have provided, which have allowed us to significantly improve both the quality of our paper. During the rebuttal and discussion stages, we have addressed all points raised. The main revisions are as follows:

* Following the suggestion of reviewer amm4, we have added more comparative information with existing flare datasets, including distinctions between indoor vs. outdoor scenes, daytime vs. nighttime conditions, and single-light-source vs. multi-light-source settings.

* As suggested by reviewer arqH, we will move more important content from the supplementary materials into the main paper.

* Based on the feedback from reviewer mFc9, we have included additional ablation studies and provided more detailed inference implementation details.

* Under the guidance of reviewer 5ETG, we have further explored the relationship between internal optical systems and flare formation, clarifying our advantages over previous works. We have also added more baseline comparison results.

These revisions have been unanimously acknowledged by all reviewers, who chose to either maintain positive scores or raise them. We hope that our joint efforts will advance the field of flare removal. We sincerely thank you for your contributions and efforts.

Best regards,

Authors of Paper 1188

---

### Decision · Program_Chairs · 2025-09-18

**Decision:**

Accept (poster)

**Comment:**

The paper presents FlareX, a physics-informed dataset for lens flare removal via combined 2D synthesis and 3D rendering. It provides large-scale templates and rendered pairs, along with a masking-based real test set. Strengths include strong motivation, clear presentation, and consistent improvements across benchmarks. Weaknesses are the small real-world test set, reliance on monocular depth, and limited details on optical modeling. Reviewer scores: 3 (borderline accept, leaning positive) + 1(accept). After rebuttal, concerns were largely addressed and reviewers maintained or raised scores. Therefore, I decide to accept this paper.